# Limited Mandibular Movements as a Consequence of Unilateral or Asymmetrical Temporomandibular Joint Involvement in Juvenile Idiopathic Arthritis Patients

**DOI:** 10.3390/jcm9082576

**Published:** 2020-08-08

**Authors:** Tamara Pawlaczyk-Kamieńska, Tomasz Kulczyk, Elżbieta Pawlaczyk-Wróblewska, Maria Borysewicz-Lewicka, Marek Niedziela

**Affiliations:** 1Department of Risk Group Dentistry, Chair of Pediatric Dentistry, Poznan University of Medical Sciences, 61-701 Poznań, Poland; klstomdz@ump.edu.pl; 2Section of Dental Radiology, Department of Biomaterials and Experimental Dentistry, Poznan University of Medical Sciences, 61-701 Poznań, Poland; tkulczyk@ump.edu.pl; 3Department of Pediatric Endocrinology and Rheumatology, Poznan University of Medical Sciences, 61-701 Poznań, Poland; ewroblewska@ump.edu.pl (E.P.-W.); mniedzie@ump.edu.pl (M.N.)

**Keywords:** juvenile idiopathic arthritis, temporomandibular joint, mandibular movements, facial asymmetry

## Abstract

This study aimed to assess the asymmetry of the lower face and motor dysfunction of the masticatory system resulting from unilateral or asymmetrical bilateral temporomandibular joint (TMJ) involvement in juvenile idiopathic arthritis (JIA) patients. The study consisted of clinical examination and cone beam computed tomography (CBCT) of TMJs. Statistical analysis showed correlations between several factors: facial asymmetry and mandibular lateral deviation; the direction of mandibular deviation and the degree of radiological deformities in TMJs; the child’s age at the time of the onset and the range of lateral movement towards the healthy or less destructed joint. In addition, there was a significant difference in ranges of lateral movements; a significantly smaller range was observed for the joint with fewer condylar abnormalities compared to the range in the opposite direction. In JIA children, among the clinical markers of unilateral or asymmetrical TMJ involvement, the asymmetry of the lower face, deviation of the mandible on opening, and an uneven range of mandibular lateral movements deserve attention. The obtained results do not show a relationship between the degree of condylar changes and the asymmetry of the lower face and the presence and degree of mandibular motor dysfunction.

## 1. Introduction

### 1.1. Background

Juvenile idiopathic arthritis (JIA) is the most common systemic inflammatory disease of connective tissue during developmental ages [1]. It begins before the age of 16 and persists for at least six months. Throughout the disease, immune-mediated inflammatory changes affect synovial membranes. Temporomandibular joint (TMJ) involvement often co-occurs with the involvement of other synovial joints, but it is occasionally the sole location of the disease [2,3,4,5]. Unfortunately, due to an absence or sparsity of clinical symptoms, this joint is often overlooked, hence its nickname, the ‘forgotten joint’ [5]. It must be stressed that TMJ is characterized by adaptive growth, and even if the joint is affected, JIA patients often do not show symptoms and signs of TMJ dysfunction [4,5,6]. TMJ arthritis, by damaging the intra-articular growth site of the condylar cartilage, can contribute to the inhibition of mandibular growth and development, leading to dentofacial deformity [7]. This dysmorphic development in JIA patients is a consequence of condylar growth disturbances rather than arthritis-induced condylar damage. However, all the symptoms are late manifestations of the TMJ arthritis, associated with irreversible destruction of the condylar process [7,8,9,10,11]. The degree to which irregularities arise is closely related to the growth potential of the mandible and depends on the age of the child at the onset of the disease, as well as on its type and course (characterized by the periods of exacerbation and remission) [4]. The dentofacial development and TMJ dysfunction also depend on the number of joints involved and the degree of destruction occurring in each of them. Unilateral (or more advanced in one of them) joint involvement includes abnormal mandibular osteogenesis affecting one side and, consequently, may result in facial asymmetry and/or mandibular deviation during maximum mouth opening (MMO) [2,4,8,9,10,11,12,13]. Unilateral TMJ arthritis is observed in 17–50% of JIA patients. However, the disease may begin in just one of the joints, while the other becomes involved with time [5,7,8,14,15,16,17,18,19]. Orofacial consequences of TMJ involvement are very diverse—from clinically asymptomatic and visualizable only by magnetic resonance imaging (MRI), to dysfunctions and/or morphological deformations that can adversely affect the quality of life [20].

### 1.2. The Aim of the Study

This study aimed to assess the symmetry of the lower face and motor dysfunction of the masticatory system resulting from unilateral or asymmetrical bilateral TMJ involvement in JIA patients.

## 2. Material and Methods

Studies involving patients of the Department of Pediatric Endocrinology and Rheumatology, Poznan University of Medical Sciences, suffering from juvenile idiopathic arthritis were conducted as part of a research program approved by the Ethical Committee of the Poznan University of Medical Sciences, Poland (No 1255/18). Inclusion criteria were (1) the diagnosis of JIA according to the criteria outlined by the International League of Association for Rheumatology (ILAR) [1] and (2) unilateral condylar deformity or bilateral condylar changes with different forms of abnormalities diagnosed via cone beam computed tomography (CBCT). A radiological grading system by Billiau et al. (2007) [21] was modified to assess the morphology of the condylar process. It describes radiologically normal appearance (score 0), cortical bony erosions (score 1), flattening (score 2), condylar flattening with additional erosions (score 3), or a complete loss of the condyle (score 4) (Figure 1). Each of the two joints were assessed independently, and one score was given for the right joint and one score for the left one. Exclusion criteria were (1) history of craniofacial surgery, (2) coexisting genetic diseases, (3) history of craniofacial injury, (4) CBCT images of poor quality. The methodology of patients’ selection is presented in Figure 2. Data on the systemic aspects of the disease were obtained from the records of the Rheumatological Clinic, after receiving the consent of the patients’ legal guardians. After presenting the study procedure, legal guardians of young patients granted written consent to their participation.

Dental clinical and CBCT examination were carried out in the same day. One dentist, specialist in dental prosthetics (TP-K), blinded to the results of the CBCT, carried out the clinical examination in a dental office. The following assessments were done:

### 2.1. Range of MMO 

A distance between two anatomical landmarks: incision superius and incision inferius (Table 1), measured with a ruler. According to standardized research techniques [22], the distance was measured twice. The reduced range of MMO was assumed to be <35 mm for children under ten years old, and <40 mm for children over ten years old [23].

### 2.2. Mandibular Lateral Deviation During MMO

For this purpose, in the centric occlusion and at the MMO, a line was drawn with a pencil on the lower incisors as an extension of the line between the upper central incisors. The distance between these lines (on lower incisors) showed the degree of deviation from the midline. The lateral mandibular deviation was reported if the deviation was ≥2 mm [24].

### 2.3. Range of Lateral Movements

For this purpose, in the centric occlusion, a line was drawn with a pencil on the lower incisors as an extension of the line between the upper central incisors. Then, in the final phase of the occlusal lateral movements (right or left), the distance between these lines was measured. A reduced range of lateral movement was reported if the distance was <7 mm [24].

### 2.4. Maximum Mandible Protrusion

This was measured as the distance between the labial surface of the upper incisors and the lingual surface of the lower incisors at maximal protrusion. A reduced range was reported if the distance was <4 mm.

### 2.5. Facial Asymmetry

This examination consisted of a direct visual assessment of the patient’s face and the analysis of a picture taken during the examination. Odd facial landmarks (trichion, ophryon, pronasale, subnasale, stomion, gnathion) (Table 1) forming the midline were marked on the picture and assessed for the extent to which they lie in a straight line, dividing the face into two halves.

### 2.6. CBCT Examination

CBCT examinations (Scanora 3D XL, Soredex Finland, Tuusula, Finland) were performed in an upright position with closed jaws. Radiographic data were reconstructed into corrected sagittal and corrected axial images of condyles (OnDemand 3D, CyberMed, Seoul, Korea) for further evaluation. The CBCT scans of both TMJs of all 39 patients were analyzed by a dentist, specialist of dental surgery and specializing in the assessment of CBCT scans (TK). To assess intra-rater reliability, 10 CBCT scans of TMJs were randomly selected and analyzed twice in two weeks interval. The intra-rater reliability was found to be good (κ= 0.85). Radiological evaluation was blinded to the clinical data. Based on the CBCT scans all of the 39 patients were classified, taking into account the grading system by Billiau et al. (2007) [21] for each of two TMJs, to one of the subgroup: JIA 0-0, JIA 0-1, JIA 0-2, JIA 0-3, JIA 0-4, JIA 1-1, JIA 1-2, JIA 1-3, JIA 1-4, JIA 2-2, JIA 2-3, JIA 2-4, JIA 3-3, JIA 3-4, JIA 4-4 [25]. In compliance with the inclusion criteria of these study patients with unilateral or bilateral with different scores of the index on both condylar process (when one of the two sides TMJ was affected more severely concerning the other one), joint involvement was qualified for further analysis. To the study were qualified the patients with at least one joint involvement (JIA 0-1, JIA 0-2, JIA 0-3, JIA 0-4) or with different degree of scores in both joints (JIA 1-2, JIA 1-3, JIA 1-4, JIA 2-3, JIA 2-4).

### 2.7. Statistical Analysis

Statistical analysis was performed using the version 12 Statistica program (StatSoft, Inc., Tulsa, OK, United States). The qualitative variables’ values are presented in multi-division tables containing the number of cases and their percentage for a given feature. The comparisons between the individual groups were performed using the Pearson χ2 nonparametric test. To determine the relationship between two measurable variables, a nonparametric Spearman rank correlation coefficient was calculated, in which r = 0 means no correlation, r ≤ 0.3 means indistinct correlation, 0.3 < r ≤ 0.5 means correlation, and r > 0.5 means strong correlation. The comparison of quantitative variable values (range of lateral movement towards the healthy side and the range of this movement towards the affected side) was performed using the Mann–Whitney test. The results were considered statistically significant when the significance level was *p* ≤ 0.05.

## 3. Results

Thirty-nine patients entered the study. Three patients were excluded from further analysis due to poor quality CBCT results for one or both TMJ (resulting from severe motion artefact). Condylar deformations were found in 21 (58.33%) patients, of which seven were excluded from further analysis due to bilateral (with the same scores in both joints: JIA 1-1, JIA 2-2-, JIA 3-3) radiological TMJ abnormalities. Finally, fourteen patients aged 12.07 ± 3.89 years met the inclusion criteria (Figure 2). In 7 children, aged 12.28 ± 4.31 years old, was diagnosed unilateral TMJ involvement, of which 4 (11.50 ± 3.87 years old) was qualified to subgroup JIA 0-1 and 3 (13.33 ± 5.5 years old) to subgroup JIA 0-3. Another seven JIA children, aged 11.86 ± 3.76 years old, had bilateral changes. Two of them (12.00 ± 7.07 years old) were qualified to subgroup JIA 1-2; 3 children (11.25 ± 3.01 years old) to JIA 1-3; and 1 (14 years old) to JIA 2-3. No children had a radiographic score deformation of 4. Due to a small number of children in particular subgroups, no statistical analysis was performed between them.

The medical characteristic of JIA patients included in the study is presented in Table 2. Sixty-four per cent of patients were diagnosed with polyarticular arthritis (involvement of more than 4 joints), and the remaining 36% with oligoarticular arthritis (involvement of up to 4 joints). In 15.38% of patients, the course of the disease was aggressive. 

The results of the clinical study are presented in Table 3. Direct visual assessment of the patient’s face and the analysis of a picture of the patient’s face showed that facial asymmetry was observed nearly in one-fourth of the examined patients. In more than half of children, the MMO was reduced, and almost in half of examined lateral mandibular deviation (≥2 mm) on MMO was reported. The range of lateral movement to the more severely affected joint was reduced in 23%, to the less severely affected joint in 38% of patients, and the range of maximum protrusion was decreased in 23% of examined children. 

Statistical analysis showed the relationship between the presence of facial asymmetry and presence of mandible deviation on MMO as well as between the presence of symmetrical face and an undisturbed path of mandibular abduction (*p* = 0.03). For 70% of patients, who had a symmetrical face, mandibular abduction was straight (without deviations of more than 2 mm), and in all subjects who showed a deviation on MMO, face was asymmetrical. We have also found that the side of asymmetry and deviation on MOO related to the more severely affected side (*p* = 0.04). In addition, a very strong positive correlation was found between the range of MMO and the range of maximum mandibular protrusion (*p* = 0.009). 

The analysis showed a significantly smaller range towards the joint with lower condylar damage (6.08 ± 2.71 mm) compared to the range in the opposite direction (9.00 ± 2.67 mm) (*p* = 0.01) (Figure 3). Additionally, there was a very strong positive correlation between the child’s age at onset of the disease and the range of lateral movement towards the joint with lower condylar damage (*p* = 0.009). However, no such relationship was noted regarding the disease duration (*p* = 0.3).

Finally, statistical analysis showed no correlation between the degree of condylar deformity and lower face asymmetry and between the degree of condylar deformity and mandibular mobility dysfunction (except direction on deviation on MMO). In some patients, CBCT-diagnosed severe condylar abnormality does not cause significant clinical signs and symptoms. In contrast, in others, facial asymmetry and/or impaired jaw movement was noted with even slight condylar damage.

## 4. Discussion

TMJ arthritis in JIA patients may be unilateral; however, it is believed that it may begin in only one but later involve the other [5,8,9,14,15,16,18,19,26]. In the studies conducted, half of the patients had radiological TMJ abnormalities of only one joint. It should be emphasized, however, that CBCT examination allowed the precise analysis of condylar morphology, but it did not enable the diagnosis of soft tissue inflammation. The abnormalities visible in the radiological examinations result from irregularities within the bone tissue and indicate the chronic nature of the disease [26,27]. The gold standard in TMJ diagnostics in JIA patients is MRI, with which it is possible to visualize not only chronic deformations resulting from a long-term degenerative process but also active inflammation of TMJ, i.e., synovitis [27,28,29,30].

TMJ arthritis may cause damage to articular structures in addition to affecting the mandibular growth center, located below the thin cartilage layer of the articular head of the condylar process, thus contributing to the inhibition of mandibular growth [4,27]. The growth center is responsible for the growth of the entire mandible (body, ramus, and angle). Growth of the anteroposterior mandibular body involves bone remodeling, i.e., resorption on the edge of the ramus and, simultaneously, stratification on its posterior edge, thus causing distal extension of the mandible. Subsequently, as a result of bone accumulation on the base of the mandibular body, the mandible grows downwards. At the same time, the ramus lengthens due to the bone growth within the condylar process [27,29,31]. Unilateral TMJ involvement (or a more advanced form in one of the joints) may cause uneven growth of the right and left side of the jaw and cause ramus shortening as well as reducing the size mandibular body on the damaged side [5,28]. Consequently, this may lead to increased facial asymmetry [4,9,10,11,12,13,14,15,25,26,29]. In research by Keller et al. (2015) [10], facial asymmetry, assessed independently by a rheumatologist and an orthodontist, was recorded in 37% and 41% of 76 examined JIA patients aged 1.9–18.6 years old, respectively. In our study, facial asymmetry was noted in 23% of patients. It should be noted, however, that the human face is generally bilaterally symmetrical, but that symmetry is never perfect. Research by Liukkonen et al. [32] showed asymmetry of the jaw to be observable in generally healthy children.

Many researchers have shown that, apart from facial asymmetry, a clinical symptom that may suggest unilateral TMJ involvement is the deviation of the jaw on MMO [4,9,11,12,13,14,16]. Weiss et al. (2008) [14] and Hu et al. (2009) [12], examining 32 JIA patients aged 1.5–17.2 years old and 100 patients aged 1.7–19.4 years old, respectively, noted a deviation of the jaw on MMO in about 20% of all patients. However, Stoll et al. (2012) [33], who diagnosed TMJ arthritis in 45% of JIA patients, noted a significantly higher percentage of patients (49%) with TMJ arthritis who exhibited a deviation of the jaw compared to the percentage of patients (12%) who, despite having healthy TMJs, showed the deviation. Koss et al. (2014) [11] compared the frequency of this disorder between JIA and healthy children. In each group, 134 patients were examined. Statistically, in a significantly higher percentage of JIA patients (62%) compared to healthy controls (16%), mandibular deviation was reported. In our study, a deviation of the mandible exceeding 2 mm was noted in 46.15% of patients. The correlation between the direction of gnathion deviation and the direction of lower jaw deviation towards the more affected joint, confirmed by previous studies, could perhaps serve as a clinical indicator showing unilateral (or unilaterally more intense) joint involvement and inhibition of mandibular growth on the damaged side.

Researchers’ opinions on MMO of JIA patients vary. Many suggest that a clinical predictor possibly indicative of TMJ involvement is a reduced mouth opening range [14,33]. However, research by Stoll et al. (2012) [33] and Koss et al. (2014) [11] does not confirm these reports. Stoll et al. (2012) [33] did not note a significant difference between the maximum opening of JIA patients diagnosed with TMJ arthritis (33%) and that of those JIA patients whose TMJ was not inflamed (21%).

There are no reports yet published regarding the ranges of extracentric movements (lateral and protrusive movements) exhibited by JIA patients. In the literature, however, the results of studies conducted among healthy children are available. Hirsch et al. (2006) [34], when assessing the range of mandibular movement in over 1000 German children aged 10–17 years, noted the average value of lateral movement to the right as 10.2 ± 2.2 mm, to the left as 10.6 ± 2.3 mm, and protrusive movement as 8.2 ± 2.5 mm. Steinmassl et al. (2017) [35], assessing the same parameters in 146 Austrian children aged 8–10 years, noted 10.0 ± 1.8, 10.1 ± 1.9, and 9.1 ± 2.0 mm, respectively. In our studies, the range of these movements was smaller in JIA patients compared to those of the studies mentioned above. Moreover, a significantly lower range of lateral movement towards the healthy side (or less affected) was reported, compared to the range of lateral movement towards the opposite side. Additionally, there was a very strong correlation between the range of maximal mouth opening range and the range of maximal protrusion. These findings could indicate that, in this group of patients, the range of extra-central occlusal motions may be limited.

No correlation between the degree of condylar deformity and lower face asymmetry; and between the degree of condylar deformity and TMJ dysfunction confirms the study by Stoustrup et al. (2018) [25]. It needs to be highlighted that for the entire growth and development of mandible are not responsible only condylar growth centers, but the process involves complex of remodeling changes, and occurs also with growth with the alveolar process, and with subperiosteal bone apposition and bone resorption. The mandibular growth depends, among others, on maxillofacial morphology, maxillary growth, occlusal relationships, muscular activity, and orofacial functions. The process is adaptive [6], and is regulated not only by genetic but also by environmental factors. Genetically determined malocclusion may contribute to a more pronounced deformation of the lower face in JIA patients. Moreover, the degree of deformation is closely related to the growth potential of the mandible and depends on the child’s age at the onset of the disease [4,27]. The most intensive growth phase of the mandible takes place up until six years of age. Further development lasts physiologically up until the age of 21, during which the mandibular growth potential falls to 15% [4,18,27]. Our study did not show a correlation between the child’s age when diagnosed with the disease and the presence of asymmetry. Still, a very strong relationship was found between the age at the onset of the disease and the range of lateral movement towards the healthy side (with a lower index). It may suggest that the later TMJs are affected, the less limited the mandibular movements towards the healthy side will be. It can, therefore, be hypothesized that the earlier the onset of TMJ arthritis in a child, the higher the chance that the child will develop abnormalities in mandibular movements.

Based on our research and the reports of other authors, it can be assumed that clinical examination of the masticatory system in JIA patients presents low sensitivity and specificity [4,10,16,19,25,36,37,38]. However, it appears that when TMJ is affected unilaterally, asymmetry of the lower face, deviation of the mandible during abduction, and limited range of lateral movements (towards the healthy side) may be characteristic manifestations. It should be emphasized, however, dentofacial deformity and TMJ dysfunction, which are consequences of condylar growth disturbances rather than arthritis-induced condylar damage alone, do not only depend on the course of TMJ arthritis and damage to the growth center, but are also related to a complex of multiple factors impairing craniofacial development [6,37]. These factors include degenerations and deformations, mechanical overloads, dysfunctions and parafunction of the masticatory organ, as well as factors compensating for the abnormal development of the dentofacial complex [6,35,37] that TMJ arthritis directly affects through damage to the growth center and inhibition of mandibular growth, including inhibition and hypoplasia of the condylar process. This, in turn, induces TMJ deformation and dysfunction, leading to increased friction in the joint, mechanical stress, and excessive load on joint surfaces. If the adaptability of joint structures is exhausted by pathological mechanisms, deformation and dysfunction become permanent, which may result in progressive dentofacial deformity, even after the inflammation has resolved. Consequently, the resulting morphological and functional abnormalities of TMJ may contribute towards the further destruction of the joint structures. In contrast, when morphological damage to the joint is due to TMJ arthritis, but it functions optimally and is loaded correctly, the development of the masticatory organ can progress smoothly after the inflammation subsides (if there is no damage to the growth center) [6,37]. Therefore, TMJ arthritis will not always lead to orofacial disorders, dysfunctions, and/or joint destruction.

## 5. Conclusions

(1)In JIA children, unilateral or bilateral (with one of the two sides, TMJ is affected more severely concerning the other one) TMJ involvement, the asymmetry of the lower face, deviation of the mandible on maximum mouth opening, and reduced range of mandibular lateral movements deserve particular attention. These patients should be under control for dentofacial assessment development and TMJ function.(2)The obtained results do not show a relationship between the degree of condylar process deformity and the presence of asymmetry of the lower face and the presence and degree of mandibular motor dysfunction.(3)In JIA patients with severe condylar process destruction, there are no significant detectable clinically abnormalities; and facial asymmetry and/or impaired jaw movement may be noted with even slight damage to the condylar process.

## Figures and Tables

**Figure 1 jcm-09-02576-f001:**
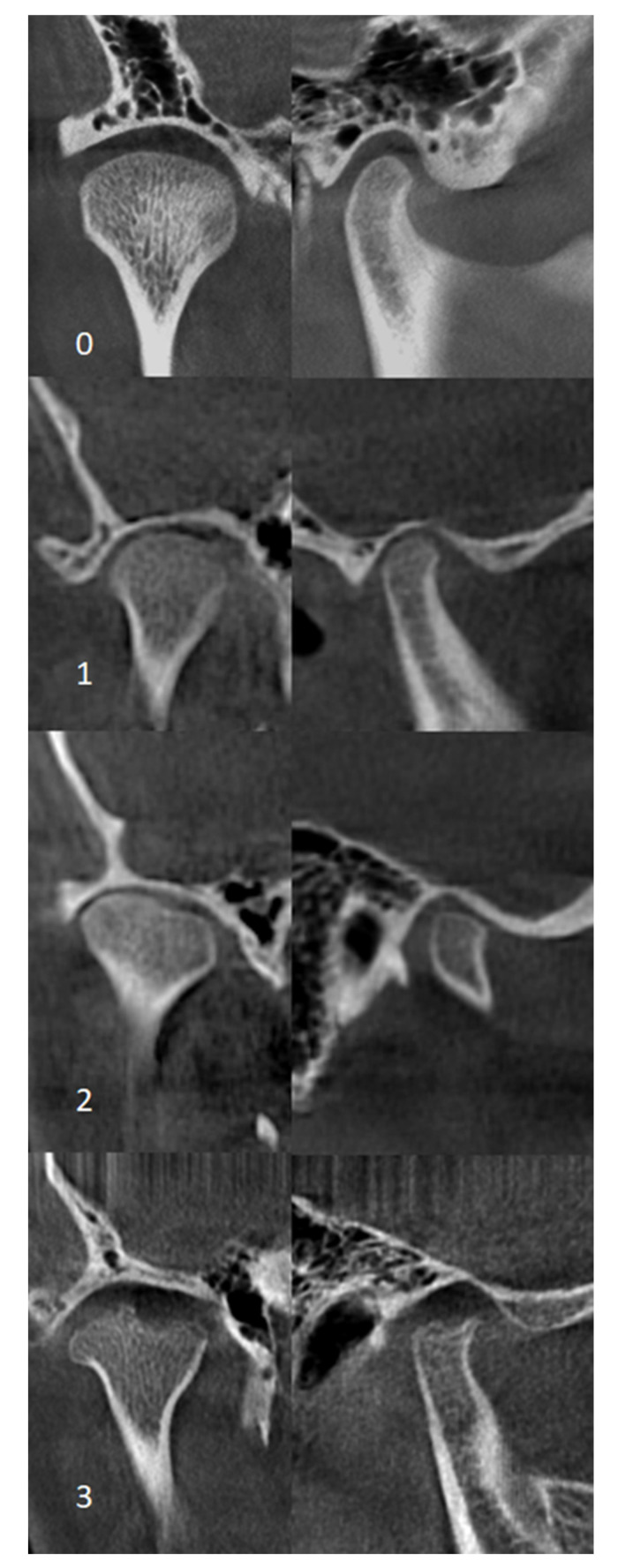
Coronal (left) and sagittal (right) cone beam computed tomography (CBCT) images for grading condylar morphology. 0—normal appearance, 1—cortical bony erosions, 2—flattening, 3—flattening with additional erosions.

**Figure 2 jcm-09-02576-f002:**
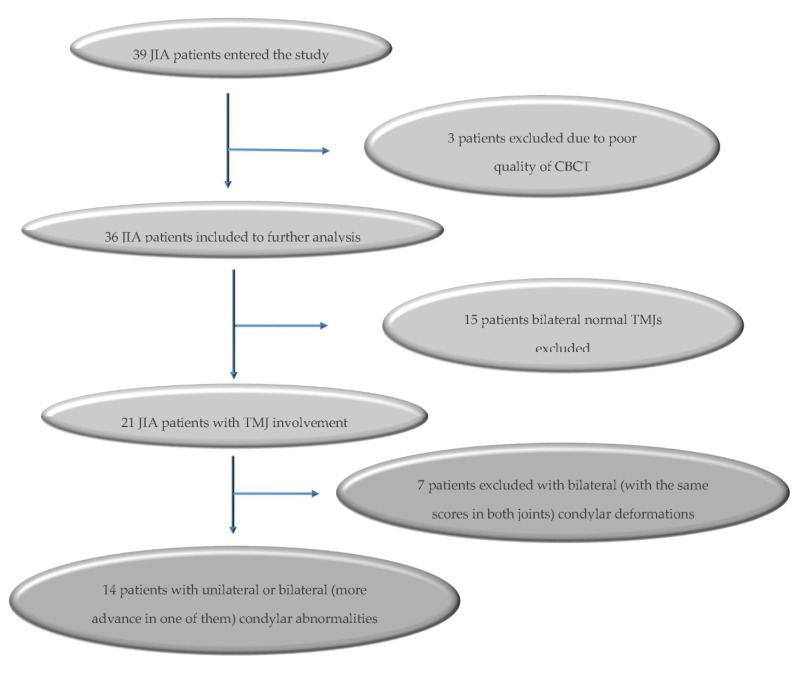
Flow chart of included juvenile idiopathic arthritis (JIA) patients.

**Figure 3 jcm-09-02576-f003:**
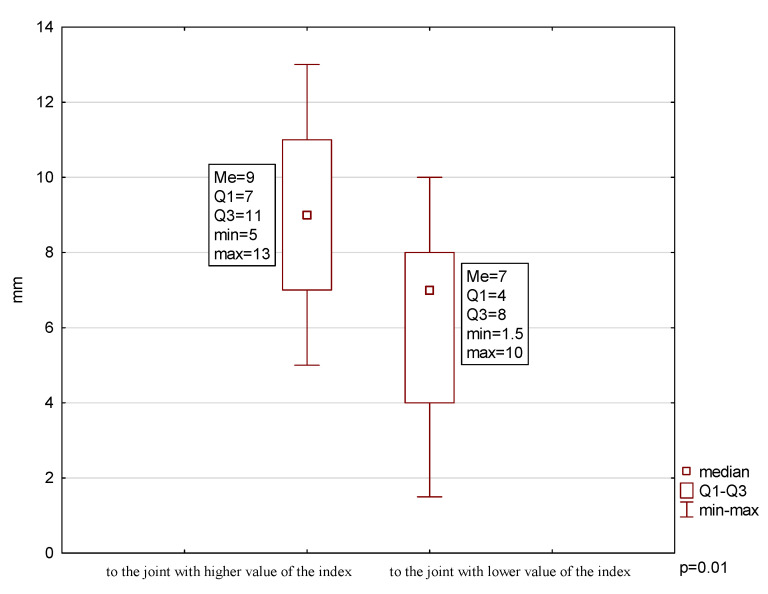
Range of mandibular lateral movements.

**Table 1 jcm-09-02576-t001:** Anatomical landmarks of face and oral cavity used in the study.

Landmark	Description
Trichion	The point on the hairline in the midline of the forehead.
Ophyron	The point at the mid-plane of a line tangent to the upper limits of the eyebrows.
Pronasale	The most protruded point of the apex nose identified in a lateral view of the rest position of the head.
Subnasale	The point at which the nasal septum merges, in the mid-sagittal plane, with the upper lip.
Stomion	The median point of the mouth when the mouth is closed.
Gnathion	The lowest median landmark on the lower border of the mandible.
Incision superius	The incisal tip of the most labially placed maxillary incisor.
Incision inferius	The incisal tip of the most labially placed mandibular incisor.

**Table 2 jcm-09-02576-t002:** Clinical characteristic of juvenile idiopathic arthritis patients included in the study.

		Number of Patients	Mean ± SD (years)	Mix (years)	Max (years)	
**Age at Examination**			11.92 ± 4.01	7	17	
**Age at Onset of the Disease**			8.31 ± 3.9	3	15	
**JIA Duration at Time of Study Entry**			3.62 ± 1.76	1	8	
**JIA Subcategories**	polyarticular	9				
	oligoarticular	5				
**Medication**	NSAIDs	6				
	DMARDs	5				
	biologic	3				

Abbreviation: NSAIDs: non-steroidal anti-inflammatory drugs; DMARDs: disease-modifying anti-rheumatic drugs.

**Table 3 jcm-09-02576-t003:** Orofacial characteristic of juvenile arthritis patients included in the study.

	Facial Asymmetry	Range of Mandibular Movement				Mandibular Lateral Deviation on MMO
		MMO	Laterotrusion to the Joint with the Higher Value of the Index	Laterotrusion to the Joint with the Lower Value of the Index	Protrusion	
% patients	23.08	↓54.85	↓23.08	↓38.46	↓23.08	46.15
Mean ± SD (mm)	-	37 ± 9.1	9.00 ± 2.67	6.08 ± 2.71	4.00 ± 2.03	-
Min (mm)	-	20	5	1.5	1	-
Max (mm)	-	48	13	10	8	-

Abbreviation: MMO: maximum mouth opening, ↓: decrease.

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
