# Peer review of "Limited Mandibular Movements as a Consequence of Unilateral or Asymmetrical Temporomandibular Joint Involvement in Juvenile Idiopathic Arthritis Patients"

_jcm, 2020, doi:10.3390/jcm9082576_

Round 1
Reviewer 1 Report
The article entitled "LIMITED MANDIBULAR MOVEMENTS AS A CONSEQUENCE OF UNILATERAL OR ASYMMETRICAL TEMPOROMANDIBULAR JOINT INVOLVEMENT IN JUVENILE IDIOPATHIC ARTHRITIS PATIENTS" regards the functional outcome of the temporomandibular joint involvement in patients with juvenile idiopathic arthritis. The manuscript needs improvements.
Introduction: try focus on the mandibular functions and dysfunctions due to the temporomandibualr joint involvement and how the dysfunctions themselves influence the altered morphology of the condyles during growing, worsening in turn the effects of the juvenile idiopatic arthritis. In fact the temporo-mandibular joint is characterized by an "adaptive type" of growth (Acta Odont Scand. 1976;34:117-126)
Methods: the study was conducted on 14 patients, not 39; please add a flow chart, explain the selection in the Methods section and report the mean age of the patient group with the SD. Also add a table with the informations about the diagnosis and therapy of the selected patients.
Central occlusion is not a scientific term.
Please refer how did you calculate the asymmetry of the condyles, especially the different height between sides that is an important data.
The functional data are clinically detected only; to improve objectivity and the scientific evaluation of the study, instrumental recordings compared and correlated to clinical findings are needed.
Results: the results are not clearly described and table 2 is hardly comprehensible. Please try to improve the manuscript reporting the results in a simple and coordinate way.
Discussion: same comment, try to present the results in a logical sequence and give an explanation remembering that the growth of the temporomandibular joint is adaptive, that inflammation influences function, and the function in turn conditions the morphology of the temporomandibular joint. It is true that the clinical functional evaluation is of great importance, but the study needs a more objective approach.
Author Response
x
Reviewer #1
We would like to thank the reviewer for the careful and thorough reading of this manuscript and invaluable comment. We appreciate the positive feedback from the reviewer. As suggested by the reviewer, we have reviewed carefully the manuscript. The following is our response (the reviewer’s comment is in italics):
The article entitled "LIMITED MANDIBULAR MOVEMENTS AS A CONSEQUENCE OF UNILATERAL OR ASYMMETRICAL TEMPOROMANDIBULAR JOINT INVOLVEMENT IN JUVENILE IDIOPATHIC ARTHRITIS PATIENTS" regards the functional outcome of the temporomandibular joint involvement in patients with juvenile idiopathic arthritis. The manuscript needs improvements.
Introduction: try focus on the mandibular functions and dysfunctions due to the temporomandibualr joint involvement and how the dysfunctions themselves influence the altered morphology of the condyles during growing, worsening in turn the effects of the juvenile idiopatic arthritis. In fact the temporo-mandibular joint is characterized by an "adaptive type" of growth (Acta Odont Scand. 1976;34:117-126)
Response:
We fully agree with the reviewers comment. Thank you very much for pointing out this problem. Thank you very much for the article you have mention. We have found the papers very interesting and we have decided to cite this publication. In the revised paper we have added the following sentences (changes are in red).:
Unfortunately, due to an absence or sparsity of clinical symptoms, this joint is often overlooked, hence its nickname, the ‘forgotten joint’ [5]. It must be stressed that TMJ is characterized by adaptive growth, and in JIA patients TMJ dysfunction due to TMJ arthritis is very rare [6]. TMJ arthritis, by damaging the intra-articular growth site of the condylar cartilage, can contribute to the inhibition of mandibular growth and development, leading to dentofacial deformity [7]. This dysmorphic development in JIA patients is a consequence of condylar growth disturbances rather than arthritis-induced condylar damage. However, these symptoms are late manifestations of the TMJ arthritis, associated with irreversible damage of the condylar process [7–11].
Also in the discussion section is more about the influence of juvenile arthritis (TMJ arthritis) on TMJ function. The results of our study showed, that there was no correlation between the degree of condylar process deformity and lower face asymmetry or dysfunctional mandibular mobility. We would like to apologize, but we thought that maybe it will be good to finish the manuscript with these considerations.
Methods: the study was conducted on 14 patients, not 39; please add a flow chart, explain the selection in the Methods section and report the mean age of the patient group with the SD. Also, add a table with the information's about the diagnosis and therapy of the selected patients.
Response
Thank you very much for these comments. Thank you very much for the idea to show the methodology of patients selection by a flow chart. We hope we have now clarified the patients’ inclusion criteria.
Reviewer #2 suggested us to remove the number of patients entered the study from the Methods section. But as You had suggested we have added the mean aged with the SD of all 39 patients in the Results section. We have also added a table of the clinical characteristic of juvenile idiopathic arthritis patients included in the study (in the results).
Central occlusion is not a scientific term.
Response:
We would like to apologize for this mistake. We have changed this term to centric occlusion.
Please refer how did you calculate the asymmetry of the condyles, especially the different height between sides that is an important data.
Response:
Thank you very much for pointing out this problem and apologize for not being precise enough. In the revised manuscript we have switched the word “asymmetrical”. Now the sentence is written as (changes are in red):
Based on the results of the CBCT examination, patients with unilateral or bilateral with different scores of the index on both condylar process (when one of the two sides TMJ was affected more severely with respect to the other one) joint involvement were qualified for further analysis (JIA 0-1, JIA 0-2, JIA 0-3, JIA 0-4, JIA 1-2, JIA 1-3, JIA 1-4, JIA 2-3, JIA 2-4).
Results: the results are not clearly described and table 2 is hardly comprehensible. Please try to improve the manuscript reporting the results in a simple and coordinate way.
Response:
Thank you very much for this comment. We fully agree with the reviewer. Your suggestion is very valuable. We have carefully revised the results section and rewritten this part of the manuscript. We hope that we have clarified the text.
Discussion: the same comment, try to present the results in a logical sequence and give an explanation remembering that the growth of the temporomandibular joint is adaptive, that inflammation influences function, and the function in turn conditions the morphology of the temporomandibular joint. It is true that the clinical functional evaluation is of great importance, but the study needs a more objective approach.
Response:
Thank you very much for this comment. We have carefully revised the discussion section and we have made essential changes.
Reviewer 2 Report
The manuscript under review attempts to evaluate the asymmetry of the lower face and motor dysfunction of the masticatory system resulting from unilateral or bilateral temporomandibular joint (TMJ) involvement in juvenile idiopathic arthritis (JIA) patients. The study is of sound design and of clear practical and clinical interest but some improvements are needed. I suggest to accept this article with minor revision.
1-The manuscript requires a review by and experienced english editor, to correct some spelling and grammar error.
Abstract: line 17 replace symmetry with asymmetry
4-INTRODUCTION
-In introduction section please introduce 2 subparagraph (Background and Aim) to clear up the aim of the study. -The Introduction section should be implemented please refer to some recent reference: Abate, A.; Cavagnetto, D.; Fama, A.; Matarese, M.; Bellincioni, F.; Assandri, F. Efficacy of Operculectomy in the Treatment of 145 Cases with Unerupted Second Molars: A Retrospective Case–Control Study. Dent. J. 2020, 8, 65. https://doi.org/10.3390/dj8030065 5-Materials and Methods: Line 69: You should remove " 39 " Line 74:please list the variables considered. The following assessments were done: Range of MMO;Mandibular lateral deviation during MMO;Range of lateral movements; Maximum mandible protrusion;Facial asymmetry; Line 96: replace "Facial symmetry" with "Facial asymmetry" The number of patients with unilateral and bilateral TMJ involvement should be provide in the M&M section. 6-RESULTS Please move the paragraph from line 127 to 135 of the results to the Materials and methods section after line 72. 7- DISCUSSION Discussion section fully explains the results of the study and compare them with the various studies in the literature. line 168 to 170 : Repetitive. Please remove the sentence. line 173 I suggest the authors to add the reference: Maspero C, Farronato M, Bellincioni F, Cavagnetto D, Abate A. Assessing mandibular body changes in growing subjects: a comparison of CBCT and reconstructed lateral cephalogram measurements. Sci Rep. 2020;In press. doi:10.1038/s41598-020-68562-6 8- CONCLUSION Conclusion is weak. Please try to describe better the results of the present study and their clinical relevance.
Author Response
c
Reviewer #2
We would like to thank the reviewer for the valuable comments and suggestions, which help to improve the quality of this manuscript. Please, find below our response (the reviewer’s comments are in italics).
The manuscript under review attempts to evaluate the asymmetry of the lower face and motor dysfunction of the masticatory system resulting from unilateral or bilateral temporomandibular joint (TMJ) involvement in juvenile idiopathic arthritis (JIA) patients. The study is of sound design and of clear practical and clinical interest but some improvements are needed. I suggest to accept this article with minor revision.
1-The manuscript requires a review by and experienced english editor, to correct some spelling and grammar error.
Abstract: line 17 replace symmetry with asymmetry
Response:
Thank you very much for these comments. As suggested by the reviewer this sentence has been changed. Correction of grammatical errors and English improvement were carried done by the professional editing service as suggested.
4-INTRODUCTION
In introduction section please introduce 2 subparagraph (Background and Aim) to clear up the aim of the study. -The Introduction section should be implemented please refer to some recent reference: Abate, A.; Cavagnetto, D.; Fama, A.; Matarese, M.; Bellincioni, F.; Assandri, F. Efficacy of Operculectomy in the Treatment of 145 Cases with Unerupted Second Molars: A Retrospective Case–Control Study. Dent. J. 2020, 8, 65. https://doi.org/10.3390/dj8030065
5-Materials and Methods: Line 69: You should remove " 39 " Line 74:please list the variables considered. The following assessments were done: Range of MMO;Mandibular lateral deviation during MMO;Range of lateral movements; Maximum mandible protrusion;Facial asymmetry; Line 96: replace "Facial symmetry" with "Facial asymmetry" The number of patients with unilateral and bilateral TMJ involvement should be provide in the M&M section. 6-RESULTS Please move the paragraph from line 127 to 135 of the results to the Materials and methods section after line 72.
Response:
Thank you very much for this vulnerable comments. We have carefully revised the manuscript and we have made essential changes.
7- DISCUSSION Discussion section fully explains the results of the study and compare them with the various studies in the literature. line 168 to 170 : Repetitive. Please remove the sentence. line 173 I suggest the authors to add the reference: Maspero C, Farronato M, Bellincioni F, Cavagnetto D, Abate A. Assessing mandibular body changes in growing subjects: a comparison of CBCT and reconstructed lateral cephalogram measurements. Sci Rep. 2020;In press. doi:10.1038/s41598-020-68562-6
Response:
We have removed the sentence as suggested by the reviewer.
Thank you very much for your suggestion to add to the reference the article. We have found the papers very interesting and we have decided to cite this publication.
8- CONCLUSION Conclusion is weak. Please try to describe better the results of the present study and their clinical relevance.
Response:
Thank you very much for this comment. We have carefully revised the conclusions and we have rewritten as (changes are in red)
- In JIA children, unilateral or bilateral (when one of the two sides TMJ is affected more severely with respect to the other one) TMJ involvement, the asymmetry of the lower face, deviation of the mandible on maximum mouth opening and reduced range of mandibular lateral movements deserve particular attention. These patients should be under control to for assessment dentofacial development and TMJ function.
- The obtained results do not show a relationship between the degree of condylar process deformity and the presence of asymmetry of the lower face and the presence and degree of mandibular motor dysfunction.
- In JIA patients with severe condylar process destruction, there are no significant detectable clinically abnormalities; and facial asymmetry and/or impaired jaw movement may be noted with even slight damage to the condylar process.
Reviewer 3 Report
In the paper under review the mandibular movement as a consequence of unilateral or asymmetrical temporomandibular joint involvement in juvenile arthritis patients is described.
However, it remains the new aspects and the innovative character remain unclear.
Numerous studies have been published presenting a large scope of findings in these patients.
Ferraz et a. (2012) described clinical aspects, Arvidsson et al. (2009) even presented longitudinal data and Kjellberg (1998) reported the growth of the face. The impairment of function was assessed by Olson et al in 1991 and Stabrun.
In a recent study of Cedströmer et al. (2020) condylar alterations and the facial growth has been described.
Thus, the results of the present study are not new, but a replication of former studies.
Additionally, the reliability of the assessment of the CBCT-examinations remains unclear (not intra-rater reliability assessment was performed). Furthermore, the clinical examination was done without using a standardized, internationally accepted examination protocol (e.g. DC/TMD or RDC/TMD).
With respect to the statistical assessment, a sample size calculation would have been helpful.
Author Response
Reviewer #3
We would like to thank the reviewer for the careful and thorough reading of this manuscript and invaluable comments and constructive suggestions. Please find our point-by-point responses (the reviewer’s comments are in italics).
In the paper under review the mandibular movement as a consequence of unilateral or asymmetrical temporomandibular joint involvement in juvenile arthritis patients is described.
However, it remains the new aspects and the innovative character remain unclear.
Numerous studies have been published presenting a large scope of findings in these patients.
Ferraz et a. (2012) described clinical aspects, Arvidsson et al. (2009) even presented longitudinal data and Kjellberg (1998) reported the growth of the face. The impairment of function was assessed by Olson et al in 1991 and Stabrun.
In a recent study of Cedströmer et al. (2020) condylar alterations and the facial growth has been described.
Thus, the results of the present study are not new, but a replication of former studies.
Response:
Thank you very much for these comments.
The problem of TMJ in JIA children in our opinion is very important, special that one, and maybe the most important, of the primary treatment goals in JIA young patients is to avoid unwanted dysmorphic dentofacial development and ensure optimal growth and development of the masticatory system. As we know for previous studies clinical consequence of TMJ involvement in this group of patients can be dysmorphic mandibular development, occlusal instability, alterations in muscular activity or TMJ dysfunctions.
According to our knowledge, there are no reports yet published regarding the ranges of extracentric movements of the mandible (lateral and protrusive movements) exhibited by JIA patients. Also in our study, we have found a significantly smaller range of movement towards the joint with lower condylar damage compared to the range in the opposite direction. Moreover, the direction of deviation on MMO and the direction of deviation of chin from the midline (during visual face assessment) correlated with the degree of radiological changes in both joints - the mandible deviated towards a more severely affected condylar process.
Additionally, the reliability of the assessment of the CBCT-examinations remains unclear (not intra-rater reliability assessment was performed).
Response:
We fully agree with reviewers comment and apologize for not mention this in the manuscript. Thank you very much for pointing out this problem. Your suggestion is very valuable. Of course, we have calculated intra-rater reliability using 10 CBCT scans.
We have carefully revised the method section and added the two sentences:
To assess intra-rater reliability 10 CBCT scans of TMJs were randomly selected and analyzed twice in two weeks interval. The intra-rater reliability was found to be good (κ = 0.85).
Furthermore, the clinical examination was done without using a standardized, internationally accepted examination protocol (e.g. DC/TMD or RDC/TMD).
Response:
Thank you very much for this comment. In our study, we used standardized anthropometry measurements techniques using the anatomical landmarks on head. According to standardised research techniques by Farakas, L.G. (Anthropometry of the head and face, 2nd ed.; Raven Press, New York, USA, 1994) each of the distance was measured twice.
Moreover to assess the range of mandibular movements we used Helkimo index (Helkimo, M. Studies on function and dysfunction of the masticatory system. II. Index for anamnestic and clinical dysfunction and occlusal state. Sven Tandlak Tidskr 1974, 67, 101–121.) and to assess maximum mouth opening in examined children we used ranges according to Müller et al. (Müller, L.; van Waes, H.; Langerweger, C.; Molinari, L.; Saurenmann, R.K. Maximal mouth opening capacity: percentiles for healthy children 4–17 years of age. Pediatr Rheumatol Online J 2013, 11, 17. doi.10.1186/1546-0096-11-17.)
Reviewer 4 Report
The aim of this study was to assess the symmetry of the lower face and motor dysfunction of the masticatory system resulting from unilateral or asymmetrical bilateral temporomandibular joint (TMJ) involvement in juvenile idiopathic arthritis (JIA) patients. The study consisted of clinical examination and Cone Beam Computed Tomography (CBCT) of TMJs.
The study is interesting and well written
- However, there is a lacking of some studies about
- the potential implications of orthodontic therapies in such patients after maxillofacial surgery (DOI: 10.1177/1721727X1201000208)
- the loss of occlusal support and symptoms of functional disturbances of the masticatory system (PMID: 10194735), that are not clearly described
- other rare clinical and radiological condylar pathologies (PubMed ID: 26147813)
- potential therapies with appliances (PubMed ID: 26486206)
Author Response
Reviewer #4
We sincerely thank the reviewer for constructive criticisms and comments, which were of great help in revising the manuscript. We have revised our manuscript, and we have rewritten the introductions and discussion sections. Please, find below our responses (the reviewer’s comments are in italics).
The aim of this study was to assess the symmetry of the lower face and motor dysfunction of the masticatory system resulting from unilateral or asymmetrical bilateral temporomandibular joint (TMJ) involvement in juvenile idiopathic arthritis (JIA) patients. The study consisted of clinical examination and Cone Beam Computed Tomography (CBCT) of TMJs.
The study is interesting and well written. However, there is a lacking of some studies about
- the potential implications of orthodontic therapies in such patients after maxillofacial surgery (DOI: 10.1177/1721727X1201000208)
- the loss of occlusal support and symptoms of functional disturbances of the masticatory system (PMID: 10194735), that are not clearly described
- other rare clinical and radiological condylar pathologies (PubMed ID: 26147813)
- potential therapies with appliances (PubMed ID: 26486206)
Thank you very much for your suggestion to add to the reference. We have found the papers very interesting and we have decided to cite these publications.
Round 2
Reviewer 1 Report
The manuscript has been partially improved.
Introduction: The sentence you added “…in JIA patients TMJ dysfunction due to TMJ arthritis is very rare… is unsound: it is true that often JIA children do not show symptoms or signs of TMJ dysfunction, but this does not mean that the TMJ is not affected and it is not possible to clearly separate the damage due to JIA and the consequences on condylar growth. “Clin Rheumatol. 2018 Oct;37(10):2667-2673.” The sentence “However, these symptoms are late manifestations of the TMJ arthritis, associated with irreversible damage of the condylar process” is in contradiction with the previous one.
Results: the number of patients of your research is 14, not 39. The mean age of 39 patients does not make sense. Because growth is involved, the age is very important. There might be the bias due to the fact that the consequences on facial asymmetry and mandibular movement of the TMJ pathology will appear during growing; for this reason the age of each children is important, and even more the bone age. The results are still difficult to understand.
Materials and methods: Condylar asymmetries: which are the further analysis?
Discussion: The condylar growth center is important, but it is not responsible for the entire growth of the mandible, please correct. The significance of this sentence is not clear: “No correlation between the degree of condylar process deformity and lower 291 face asymmetry and between the degree of condylar process deformity and TMJ dysfunction showed 295 that the growth and development of temporomandibular joint is adaptive [6]” could you please better explain? Your results are expected, there is not anything new; my suggestion is to make an adequate comment.
The manuscript is still very difficult to understand, especially the results section: please try to careful describe what is enhanced, when, at which age etc. There still are spelling and grammar mistakes.
Author Response
c
Reviewer #1
We also greatly appreciate the reviewers for their complimentary comments. We agree with almost all their comments, and we apologize for the errors. Please, find below our response (the reviewer’s comments are in italics).
Introduction: The sentence you added “…in JIA patients TMJ dysfunction due to TMJ arthritis is very rare… is unsound: it is true that often JIA children do not show symptoms or signs of TMJ dysfunction, but this does not mean that the TMJ is not affected and it is not possible to clearly separate the damage due to JIA and the consequences on condylar growth. “Clin Rheumatol. 2018 Oct;37(10):2667-2673.” The sentence “However, these symptoms are late manifestations of the TMJ arthritis, associated with irreversible damage of the condylar process” is in contradiction with the previous one.
Response:
We fully agree with reviewers comment and apologize for these mistakes. Thank you very much for pointing out this problem. Your suggestion is very valuable. You are absolutely right, I was meaning that symptoms are rare clinically detectable, not that TMJ dysfunctions are rare. Apologize for that. Now it is rewritten as (changes are in red).:
It must be stressed that TMJ is characterized by adaptive growth, and in even if the join is affected, JIA patients TMJ dysfunction due to TMJ arthritis is very rare often do not show symptoms and signs of TMJ dysfunction [4-6].
Results: the number of patients of your research is 14, not 39. The mean age of 39 patients does not make sense. Because growth is involved, the age is very important. There might be the bias due to the fact that the consequences on facial asymmetry and mandibular movement of the TMJ pathology will appear during growing; for this reason the age of each children is important, and even more the bone age. The results are still difficult to understand.
Response:
Thank you very much for this comment. The reviewer is right, there is not necessary to give the age of 39 children. We have removed this from the revised text. We have also made necessary changes, as suggested by the reviver. The number of patients in the study subgroups were too small to make statistical analysis, and to compare particular data (as facial asymmetry or mandibular movements) between the subgroups. In the revised manuscript we have clarified the results section.
Materials and methods: Condylar asymmetries: which are the further analysis?
Response:
Thank you very much the reviewer for this comment. We apologize for not being precise enough. In the revised manuscript we have rewritten this part as (changes are in red):
Based on the CBCT scans, all of the 39 patients were classified to one of the subgroup: JIA 0-0, JIA 0-1, JIA 0-2, JIA 0-3, JIA 0-4, JIA 1-1, JIA 1-2, JIA 1-3, JIA 1-4, JIA 2-2, JIA 2-3, JIA 2-4, JIA 3-3, JIA 3-4, JIA 4-4. Based on the results of the CBCT examination, In compliance with the inclusion criteria of this study patients with unilateral or bilateral with different scores of the index on both condylar process (when one of the two sides TMJ was affected more severely concerning the other one) joint involvement was qualified for further analysis. To the study were qualified the patients with at least one joint involvement (JIA 0-1, JIA 0-2, JIA 0-3, JIA 0-4) or with different degree of scores in both joints (JIA 1-2, JIA 1-3, JIA 1-4, JIA 2-3, JIA 2-4).
Discussion: The condylar growth center is important, but it is not responsible for the entire growth of the mandible, please correct. The significance of this sentence is not clear: “No correlation between the degree of condylar process deformity and lower 291 face asymmetry and between the degree of condylar process deformity and TMJ dysfunction showed 295 that the growth and development of temporomandibular joint is adaptive [6]” could you please better explain? Your results are expected, there is not anything new; my suggestion is to make an adequate comment.
Response:
The text has been revised as suggested and we have rewritten this part of our manuscript.
No correlation between the degree of condylar deformity and lower face asymmetry; and between the degree of condylar deformity and TMJ dysfunction showed that the growth and development of temporomandibular joint confirms study by Stoustrup et al. (2018) [25]. For the entire growth of mandible are not responsible only condylar growth centers, but the process of mandibular growth and development involves a complex of remodelling changes, and occurs also with growth with the alveolar process, and with subperiosteal bone apposition and bone resorption. The mandibular growth depends among others on maxillofacial morphology, maxillary growth, occlusal relationships, muscular activity and orofacial functions. The process is adaptive [6], and seems to be influenced by many is regulated not only by genetic but also by environmental factors.
We have also added a reference
Stoustrup et al.. No association between types of unilateral mandibular condylar abnormalities and facial asymmetry in orthopedic-treated patients with juvenile idiopathic arthritis. Am J Orthod Dentofacial Orthop 2018, 153, 214-223. doi:10.1016/j.ajodo.2017.05.037
The manuscript is still very difficult to understand, especially the results section: please try to careful describe what is enhanced, when, at which age etc. There still are spelling and grammar mistakes.
Response:
We would like to thank the reviewer for the detailed comments and suggestions for the manuscript. Apologize for being unclear. In the revised manuscript we have made necessary changes as suggested by the reviewer.
Apologize also for our English. The manuscript has been edited second time by an English-speaking native, so we hope it now matches the journal standard.b
Reviewer 3 Report
Thank you for addressing the comments.
Author Response
We would like to thank the reviewer for the careful and thorough reading of this manuscript and invaluable comment. We appreciate the positive feedback from the reviewer.